# Continual Event Extraction with Semantic Confusion Rectification

**Zitao Wang**[†]    **Xinyi Wang**[†]    **Wei Hu**[†, ‡, *]

[†] State Key Laboratory for Novel Software Technology, Nanjing University, China
[‡] National Institute of Healthcare Data Science, Nanjing University, China
ztwang.nju@gmail.com, xywang.nju@gmail.com, whu@nju.edu.cn

## Abstract

We study continual event extraction, which aims to extract incessantly emerging event information while avoiding forgetting. We observe that the semantic confusion on event types stems from the annotations of the same text being updated over time. The imbalance between event types even aggravates this issue. This paper proposes a novel continual event extraction model with semantic confusion rectification. We mark pseudo labels for each sentence to alleviate semantic confusion. We transfer pivotal knowledge between current and previous models to enhance the understanding of event types. Moreover, we encourage the model to focus on the semantics of long-tailed event types by leveraging other associated types. Experimental results show that our model outperforms state-of-the-art baselines and is proficient in imbalanced datasets.

## 1 Introduction

Event extraction (Grishman, 1997; Ahn, 2006) aims to detect event types and identify their event arguments and roles from natural language text. Given a sentence "*The Oklahoma City bombing conspirator is already serving a life term in federal prison*", an event extraction model is expected to identify "*bombing*" and "*serving*", which are the event triggers of the "*Attack*" and "*Sentence*" types, respectively. Also, the model should identify arguments and roles of corresponding event types such as "*conspirator*" and "*Oklahoma City*" are two arguments involved in "*Attack*" and as argument roles of "*Attacker*" and "*Place*", respectively.

Conventional studies (McClosky et al., 2011; Nguyen et al., 2016; Du and Cardie, 2020; Lin et al., 2020; Nguyen et al., 2021; Wang et al., 2022) model event extraction as a task to extract from the pre-defined event types and argument roles. In practice, new event types and argument roles emerge

---

[*] Corresponding author

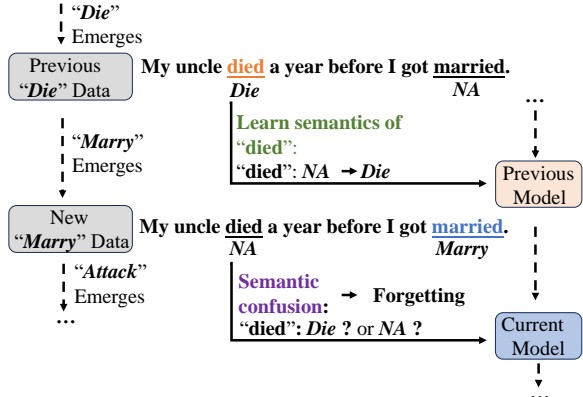

(a) Example of semantic confusion when new types emerge.

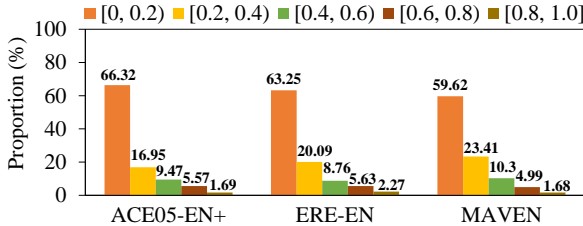

(b) Imbalanced number distribution of event types.

Figure 1: Problems in continual event extraction.

continually. We define a new problem called *continual event extraction* for this scenario. Compared to conventional studies, continual event extraction expects the model not only to detect new types and identify corresponding event arguments and roles but also to remember the learned types and roles. This scenario belongs to *continual learning* (Ring, 1994), which learns from data streams with new emerging data.

To alleviate the so-called catastrophic forgetting problem (Thrun and Mitchell, 1995; French, 1999), existing works focus on event detection and apply knowledge transfer or prompt engineering (Cao et al., 2020; Yu et al., 2021; Liu et al., 2022). On one hand, they do not consider the task of argument extraction, making them incomplete in event extraction. On the other hand, they ignore that the

semantic understanding by the model deviates from correct semantics when new types emerge, which we call *semantic confusion*.

First, semantic confusion is caused by the annotations of previous types and new types do not generate at the same time. As shown in Figure 1(a), a sentence may have multi-type annotations. However, current training data only contains new annotations, and the model misunderstands the text "*died*" with the previous annotation "*Die*" as a negative label "*NA*". Similarly, the model that is only trained on the previous data would identify the new types as negative labels. Existing works (Cao et al., 2020; Yu et al., 2021; Liu et al., 2022) simply transfer all learned knowledge to the current model, which would disturb new learning. The second problem is the imbalanced distribution of event types in natural language text. Figure 1(b) shows the number distribution of event types in three widely used event extraction datasets. The model is confused with the semantics of the long-tailed event types in two aspects. On one hand, it suffers from the lack of training on long-tailed event types due to their few instances. On the other hand, the semantics of long-tailed types would be disturbed by popular types during training.

This paper proposes a novel continual event extraction method to rectify semantic confusion and address the imbalance issue. Specifically, we propose a data augmentation strategy that marks pseudo labels of each sentence to avoid the disturbance of semantic confusion. We apply a pivotal knowledge distillation to further encourage the model to focus on vital knowledge during training at the feature and prediction levels. Moreover, we propose prototype knowledge transfer, which leverages the semantics of other associated types to enrich the semantics of long-tailed types.

Our main contributions are outlined as follows:

- Unlike existing works, we extend continual learning to event extraction and propose a new continual event extraction model.
- We explicitly consider semantic confusion on event types. We propose data augmentation with pseudo labels, pivotal knowledge distillation, and prototype knowledge transfer to rectify semantic confusion.
- We conduct extensive experiments on three benchmark datasets. The experimental results demonstrate that our model establishes a new state-of-the-art baseline with significant improvement and obtains better performance on long-tailed types.

## 2 Related Work

### 2.1 Event Extraction

Conventional event extraction models (McClosky et al., 2011; Li et al., 2013; Nguyen et al., 2016; Lin et al., 2020; Wang et al., 2021) regard the event extraction as a multi-class classification task. In recent years, several new paradigms have been proposed to model event extraction. The works (Du and Cardie, 2020; Liu et al., 2020; Li et al., 2020; Lyu et al., 2021) treat event extraction as a question-answering task. They take advantage of the pre-defined question templates and have specific knowledge transfer abilities on event types. The work (Wang et al., 2022) refines event extraction as a query-and-extract process by leveraging rich semantics of event types and argument roles. These models cannot apply to continual event extraction as they learn all event types at once.

### 2.2 Continual Learning

Mainstream continual learning methods can be distinguished into three families: regularization-based methods (Li and Hoiem, 2017; Kirkpatrick et al., 2017), dynamic architecture methods (Aljundi et al., 2017; Rosenfeld and Tsotsos, 2018; Qin et al., 2021), and memory-based methods (Lopez-Paz and Ranzato, 2017; Rebuffi et al., 2017; Castro et al., 2018; Wu et al., 2019).

For many NLP tasks, the memory-based methods (Wang et al., 2019; de Masson d'Autume et al., 2019; Cao et al., 2020) show superior performance than other methods. Existing works (Cao et al., 2020; Yu et al., 2021; Liu et al., 2022) make use of knowledge transfer to alleviate catastrophic forgetting in event detection. KCN (Cao et al., 2020) employs memory reply and hierarchical distillation to preserve old knowledge. KT (Yu et al., 2021) transfers knowledge between related types to enhance the learning of old and new event types. EMP (Liu et al., 2022) leverages soft prompts to preserve the knowledge learned from each task and transfer it to new tasks. All the above models are unable to identify event arguments and roles, so they are incomplete in continual event extraction. Furthermore, they ignore semantic confusion on event types while training. We address these problems and propose a new model.

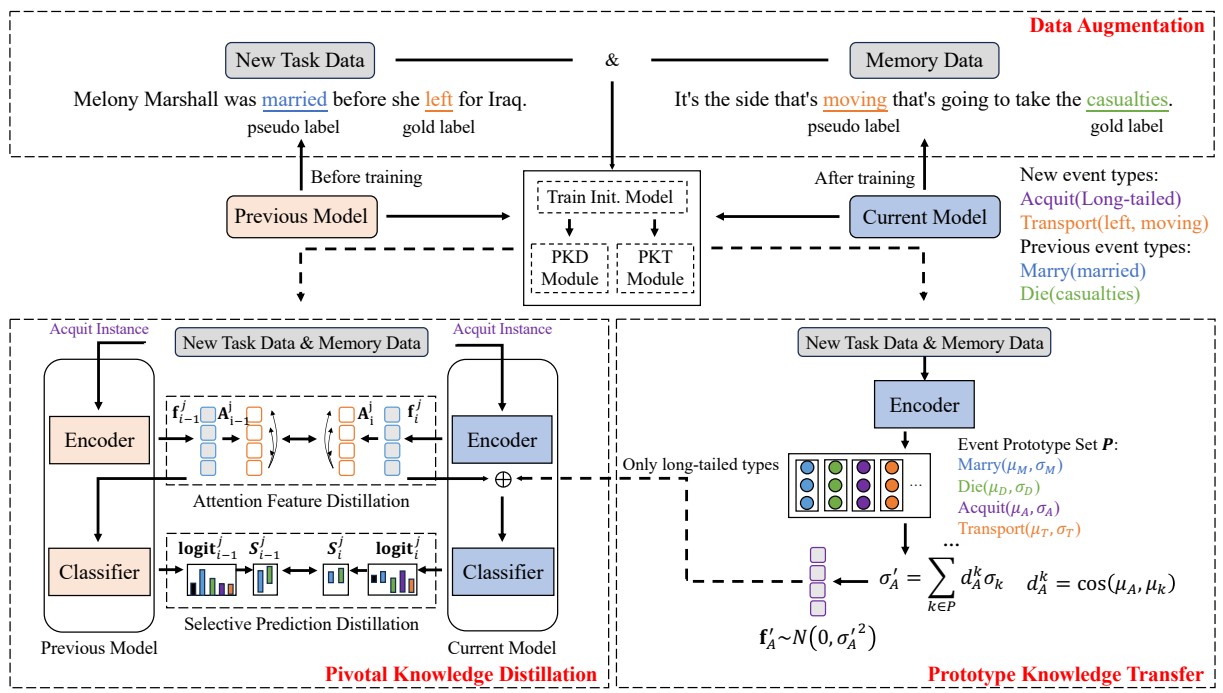

Figure 2: Framework of our proposed continued event detection model.

## 3 Task Definition

Given a sentence, the event extraction task aims to detect the event types in this sentence (a.k.a. *event detection*) and identify the event arguments and roles (a.k.a. *argument extraction*).

In a continual event extraction task, there is a sequence of $K$ tasks $\{T_1, T_2, \ldots, T_K\}$. Each individual task $T_i$ is a conventional event extraction task that contains its own event type set $E_i$, role type set $R_i$, and respective training set $D_i^{\text{train}}$, development set $D_i^{\text{dev}}$, and test set $D_i^{\text{test}}$. $E_i$ of each task $T_i$ is disjoint with other tasks. $D_i^{\text{train}}$ contains instances for $E_i$ and negative instances for "*NA*". $D_i^{\text{dev}}$ and $D_i^{\text{test}}$ only contain sentences for $E_i$.

At the $i$-th stage, the continual event extraction model is trained on $D_i^{\text{train}}$ and evaluated on all seen test sets $\tilde{D}_i^{\text{test}} = \bigcup_{t=1}^{i} D_t^{\text{test}}$ to detect all seen event types $\tilde{E}_i = \bigcup_{t=1}^{i} E_t$ and identify all event arguments and roles of corresponding event types.

## 4 Methodology

### 4.1 Overall Framework

Our framework for continual event extraction consists of two models: event detection model $\mathcal{F}_i$ and argument extraction model $\mathcal{G}_i$. When a new task $T_i$ comes, we detect the candidate event types for each sentence by $\mathcal{F}_i$. The framework of our proposed model $\mathcal{F}_i$ is shown in Figure 2. We first augment

current training data with pseudo labels. Then, we train the current model on augmented data and memory data with pivotal knowledge distillation. For long-tailed event types, we enhance their semantics with event prototypes. At last, we pick and store a few instances for new types and augment them with pseudo labels for the next task $T_{i+1}$. The parameters are updated during training. After predicting each candidate event type, we train $\mathcal{G}_i$ to obtain corresponding event arguments and roles. Similar to event detection, we also pick and store a few instances. The accuracy of argument extraction highly depends on correct event types.

### 4.2 Event Detection

#### 4.2.1 Base Model and Experience Replay

**Base model.** Following (Cao et al., 2020; Yu et al., 2021; Liu et al., 2022; Du and Cardie, 2020), we use the pre-trained language model BERT (Devlin et al., 2019) as the encoder to extract the hidden representation of text.

Given a sentence $w$, we first use the BERT encoder to get the hidden representation $\mathbf{h}_{x_j}$ for each token $x_j$ in $w$. Then, we obtain the feature representation $\mathbf{f}_{x_j}$ of $x_j$ by

$$\mathbf{f}_{x_j} = \text{LayerNorm}\big(\mathbf{W}\,\text{Dropout}(\mathbf{h}_{x_j}) + \mathbf{b}\big), \quad (1)$$

where $\mathbf{W} \in \mathbb{R}^{h \times d}$ and $\mathbf{b} \in \mathbb{R}^h$ are trainable parameters. $h, d$ are the dimensions of feature repre-

sentations and hidden layers in BERT, respectively. LayerNorm($\cdot$) is the normalization operation.

We use a linear softmax classifier to get $x_j$'s output probability distribution on the basis of $\mathbf{f}_{x_j}$. The cross-entropy classification loss of the current dataset is defined as follows:

$$\mathcal{L}_{\text{cla}} = -\frac{1}{|\mathcal{N}|} \sum_{x \in \mathcal{N}} \sum_{e \in \mathcal{E}} y_{x,e} \log P(e \mid x; \mathcal{F}_i), \quad (2)$$

where $\mathcal{N}$ is the token set from the current training data. $\mathcal{E}$ is the seen event types set $\tilde{E}_i$ and "*NA*". $y_{x,e}$ indicates whether the reference type of $x$ is $e$. $P(e \mid x; \mathcal{F}_i)$ is the probability of $x$ classified as $e$ by the event detection model $\mathcal{F}_i$.

**Experience replay.** Inspired by the previous works on continual learning (Wang et al., 2019; de Masson d'Autume et al., 2019; Yu et al., 2021; Liu et al., 2022), we pick and store a small number $m$ of instances for each event type. At the $i$-th stage, the memory space to store the current training data is denoted by $\delta_i$, so the accumulated memory space is $\tilde{\delta}_i = \bigcup_{t=1}^{i} \delta_t$. Note that we do not store negative instances in $\tilde{\delta}_i$, owing to the fact that negative instances are prevalent at each stage. At the $i$-th stage, we train the model with current training data $D_i^{\text{train}}$ and memory space $\tilde{\delta}_{i-1}$. We leverage the k-means algorithm to cluster the feature representations of each event type's instances, where the number of clusters equals the memory size $m$. We select the instances closest to the centroid of each cluster and store them.

#### 4.2.2 Data Augmentation

In the event detection task, a sentence may have several annotations of event types. For example, in the sentence "*Melony Marshall was married before she left for Iraq*", "*married*" and "*left*" indicate the event types "*Marry*" and "*Transport*", respectively. Let us assume that "*Marry*" is the previously seen type and "*Transport*" is the newly emerging type. If the current memory space does not include this sentence, the annotation corresponding to "*married*" would be "*NA*". Thus, the model would treat the text "*married*" as a negative label "*NA*" at the current stage. However, "*married*" has been considered as the event trigger of "*Marry*" at the previous stage. It is noticeable that the semantics of "*married*" are confused at these two different stages. Previous works (Cao et al., 2020; Yu et al., 2021; Liu et al., 2022) simply ignore this problem and suffer from semantic confusion.

To address this issue, we propose a data augmentation strategy with pseudo labels to excavate potential semantics and rectify semantic confusion. Before training the current model, we first augment the training data with the previous model. For each instance in the training set $D_i^{\text{train}}$, it is just annotated by the new event types $E_i$. We regard this sentence as a test instance and leverage the previous model to predict the event types for each token. Once the prediction confidence exceeds a threshold $\tau$, we mark this token as the predicted type, serving as a pseudo label. Then, the augmented data can be used to train the current model. After training, we also leverage the trained model to obtain pseudo labels for the memory data. Note that we just use augmented task data and memory data for training, rather than for prototype generation in prototype knowledge transfer, since the pseudo labels are not completely reliable.

#### 4.2.3 Pivotal Knowledge Distillation

Knowledge distillation (Hinton et al., 2015) aims to remind the current model about learned knowledge by leveraging the knowledge from the previous model. It is important to leverage precise learned knowledge, otherwise, it would lead to semantic confusion like in previous works (Cao et al., 2020; Yu et al., 2021; Liu et al., 2022). In this paper, we propose pivotal knowledge distillation, which enables the model to focus on critical knowledge and transfers precise knowledge between the previous model and the current model at the feature and prediction levels.

**Attention feature distillation.** At the feature level, we expect the features extracted by the current model similar to those by the previous model. Unlike existing works (Lin et al., 2020; Cao et al., 2020; Liu et al., 2022), we consider that each token in a sentence should not have an equal feature weight toward an event trigger. The tokens associated closely with the event trigger are more important than others. To capture such context information, we propose attention feature distillation. We first apply in-context attention to obtain attentive feature $\mathbf{A}_{x_j}$ for each token $x_j$ in a sentence:

$$\mathbf{A}_{x_j} = \frac{1}{|\mathcal{W}|} \sum_{x \in \mathcal{W}} \phi(\mathbf{f}_{x_j}, \mathbf{f}_x)\mathbf{f}_x, \quad (3)$$

where $\mathcal{W}$ denotes all tokens in this sentence. $\phi(\cdot)$ is an attention function, which is calculated as the average of the self-attention weights from the last

$L$ layers of BERT, where $L$ is a hyperparameter. $\mathbf{f}_x$ is the feature representation of $x$.

After capturing attentive feature $\mathbf{A}_{x_j}$, we preserve previous features through an attention feature distillation loss:

$$\mathcal{L}_{\text{afd}} = -\frac{1}{|\mathcal{N}|} \sum_{x \in \mathcal{N}} 1 - \cos(\mathbf{A}_x^i, \mathbf{A}_x^{i-1}), \quad (4)$$

where $\cos(\cdot)$ is the cosine similarity between two features. $\mathbf{A}_x^i$ and $\mathbf{A}_x^{i-1}$ are two attentive features computed by $\mathcal{F}_i$ and $\mathcal{F}_{i-1}$, respectively.

During feature distillation, the current model would pay more attention to the associated tokens and obtain the critical and precise knowledge of these tokens from the previous model to remember the seen event types. Moreover, with the lower attention to irrelevant tokens, the current model avoids being confused by irrelevant semantics.

**Selective prediction distillation.** At the prediction level, we enforce that the probability distribution predicted by the current model $\mathcal{F}_i$ does not deviate from that of the previous model $\mathcal{F}_{i-1}$. The previous methods (Yu et al., 2021; Song et al., 2021; Liu et al., 2022) transfer the probability distribution of each token in a sentence. However, we argue that this brings semantic confusion from the previous model. The tokens of emerging event types should not be transferred. The previous model $\mathcal{F}_{i-1}$ gives an inaccurate probability distribution of these tokens due to that it has not been trained on emerging event types. Therefore, if we transfer the current model $\mathcal{F}_i$ with a wrong probability distribution, it would be confused with the semantics of new event types. To overcome this problem, we directly leverage the tokens of the previously seen types and the "*NA*" type to transfer knowledge. Furthermore, we do not transfer the probability distribution of "*NA*" owing to the availability of negative training data on every task.

Based on the above observation, we propose a selective prediction distillation to avoid semantic confusion in knowledge distillation:

$$\mathcal{L}_{\text{spd}} = -\frac{1}{|\tilde{\mathcal{N}}|} \sum_{x \in \tilde{\mathcal{N}}} \sum_{e \in \tilde{\mathcal{E}}} P(e \,|\, x; \mathcal{F}_{i-1}) \log P(e, |\, x; \mathcal{F}_i), \quad (5)$$

where $\tilde{\mathcal{N}}$ is the token set excluding the tokens of new types. $\tilde{\mathcal{E}}$ is the previously seen type set.

Inspired by (Yu et al., 2021), we optimize the classification loss and distillation loss with multi-task learning. The final loss is

$$\mathcal{L} = \left(1 - \frac{|\tilde{E}_{i-1}|}{|\tilde{E}_i|}\right) \mathcal{L}_{\text{cls}} + \frac{|\tilde{E}_{i-1}|}{|\tilde{E}_i|}(\alpha \cdot \mathcal{L}_{\text{afd}} + \beta \cdot \mathcal{L}_{\text{spd}}), \quad (6)$$

where $\alpha$ and $\beta$ are hyperparameters.

### 4.2.4 Prototype Knowledge Transfer

The distribution of event types is naturally imbalanced in the real world, where the majority of instances belong to a few types. Due to the lack of instances, the semantics of long-tailed event types are difficult to capture by the model. Moreover, if a long-tailed event type is analogous to a popular event type with many instances, its semantics are likely to be biased toward that of the popular one. Consequently, the model obtains confused semantics on long-tailed event types.

To address this issue, we propose a prototype knowledge transfer method to enhance the semantics of long-tailed types with associated event prototypes. In our viewpoint, the prototypes imply the semantics of their event types. To get the exact prototypes of emerging event types, we first train the base model with the current training data $D_i^{\text{train}}$. For each event type $e$ in the seen event types $\tilde{E}_i$, we calculate the average $\mu_e$ and the standard deviation $\sigma_e$ of feature representations of corresponding tokens in the current training data $D_i^{\text{train}}$ or memory space $\tilde{\delta}_{i-1}$. If the event type is newly emerging, we calculate its prototype by

$$\mu_e = \frac{1}{|\mathcal{N}_e|} \sum_{x \in \mathcal{N}_e} \mathbf{f}_x, \quad (7)$$

$$\sigma_e = \sqrt{\frac{1}{|\mathcal{N}_e|} \sum_{x \in \mathcal{N}_e} (\mathbf{f}_x - \mu_e)^2}, \quad (8)$$

where $\mathcal{N}_e$ is the tokens of event type $e$ in $D_i^{\text{train}}$. For the previous event types, we compute their prototype by tokens in memory space as above.

For each token $x$ of long-tailed type $e$, we clarify its semantics through the associated event prototypes. We first measure the similarity of $e$ and another event type $e'$ in the seen event types $\tilde{E}_i$ by the cosine distance. Then, we calculate the associated standard deviation with associated event prototypes by

$$\tilde{\sigma}_e = \sum_{e' \in P} d'_e \sigma', \quad d'_e = \cos(\mu_e, \mu'), \quad (9)$$

where $P$ is the prototypes of all seen event types.

We assume that the hidden representations of event types follow the Gaussian distribution and generate the intensive vector $\tilde{\mathbf{f}}_e$ by

$$\tilde{\mathbf{f}}_e \sim \mathbb{G}(0, \tilde{\sigma}_e^2). \tag{10}$$

We add the intensive vector $\tilde{\mathbf{f}}_e$ with the feature vector $\mathbf{f}_x$ as the final representation $\mathbf{f}_x^*$ of this long-tailed token by $\mathbf{f}_x^* = \tilde{\mathbf{f}}_e + \mathbf{f}_x$. We leverage the final representation for further learning like the feature representation of popular types. Note that we do not apply the average to generate the intensive vector. We think that the use of average would align the semantics of long-tailed event types and their associated event types, causing semantic confusion to a certain extent. In this paper, we categorize the last 80% of types in terms of the number of instances as long-tail types.

## 4.3 Argument Extraction

After obtaining the event types of sentences, we further identify their arguments and roles based on the pre-defined argument roles of each event type. The arguments are usually identified from the entities in the sentence. Here, we first recognize the entities in each sentence with the BERT-BiLSTM-CRF model, which is optimized on the same current training data. We treat these entities as candidate arguments. Then, we leverage another BERT encoder to encode each candidate argument and concatenate its starting and ending token representations as the final feature vector. Finally, for each candidate event type, we classify each entity into argument roles by a linear softmax classifier. The training objective is to minimize the following cross-entropy loss function:

$$\mathcal{L}_{\text{ae}} = -\frac{1}{|\mathcal{Q}|} \sum_{e \in \mathcal{Q}} \sum_{r \in \mathcal{R}} y_{e,r} \log P(r \,|\, e), \tag{11}$$

where $\mathcal{Q}$ is the set of candidate entities. $\mathcal{R}$ is the pre-defined argument roles of the corresponding event type. $y_{e,r}$ denotes whether the reference role of $e$ is $r$. $P(r \,|\, e)$ denotes the probability of $e$ classified as $r$.

For argument extraction, we also apply another memory space to store a few instances to alleviate catastrophic forgetting. We pick and store instances in the same way as described in Section 4.2.1.

## 5 Experiments and Results

In this section, we evaluate our model and report the results. The source code is accessible online.[1]

### 5.1 Experiment Setup

**Datasets.** We carry out our experiments on 3 widely-used benchmark datasets. (1) *ACE05-EN+* (Doddington et al., 2004) is a classic event extraction dataset containing 33 event types and 22 argument roles. We follow (Lin et al., 2020) to pre-process the dataset. Since several event types are missing in the development and test sets, we re-split the dataset for a better distribution of the event types. (2) *ERE-EN* (Song et al., 2015) is another popular event extraction dataset, which contains 38 event types and 21 argument roles. We pre-process and re-split the data like ACE05-EN+. (3) *MAVEN* (Wang et al., 2020) is a large-scale dataset with 168 event types. Due to the lack of argument annotations, we can only evaluate the event detection task. Since its original dataset does not provide the annotations of the test set, we re-split the dataset as well. More details about the dataset statistics and splits can be seen in Appendix A.

**Implementation.** As defined in Section 3, we conduct a sequence of event extraction tasks. Following (Yu et al., 2021; Liu et al., 2022), we partition each dataset into 5 subsets to simulate 5 disjoint tasks, denoted by $\{T_1, \ldots, T_5\}$. As the majority of event types have more than 10 training instances, we set the memory size to 10 for all competing models. To reduce randomness, we run every experiment 6 times with different subset permutations. See Appendix B for hyperparameters.

**Competing models.** We compare our model with 7 competitors. *Fine-tuning* simply trains on the current training set without any memory. It forgets the previously learned knowledge seriously. *Joint-training* stores all previous training data as memory and trains the model on all training data for each new task. It simulates the behavior of re-training. It is viewed as an upper bound in continual event extraction. To reduce catastrophic forgetting, *KCN* (Cao et al., 2020) combines experience reply with hierarchical knowledge distillation. *KT* (Yu et al., 2021) transfers knowledge between learned event types and new event types. *EMP* (Liu et al., 2022) uses prompting methods to instruct model

---

[1]https://github.com/nju-websoft/SCR

| | Datasets | ACE05-EN+ | | | | | ERE-EN | | | | | MAVEN | | | | |
|---|---|---|---|---|---|---|---|---|---|---|---|---|---|---|---|---|
| | Models | $T_1$ | $T_2$ | $T_3$ | $T_4$ | $T_5$ | $T_1$ | $T_2$ | $T_3$ | $T_4$ | $T_5$ | $T_1$ | $T_2$ | $T_3$ | $T_4$ | $T_5$ |
| Event detection | Fine-tuning | 84.35 | 54.37 | 36.82 | 31.15 | 24.75 | 80.73 | 48.60 | 36.08 | 27.28 | 24.98 | 78.88 | 47.47 | 35.16 | 28.79 | 22.67 |
| | Joint-training | 84.35 | 81.24 | 80.02 | 77.03 | 75.51 | 80.73 | 73.61 | 69.75 | 65.78 | 63.66 | 78.88 | 71.78 | 68.52 | 67.74 | 65.88 |
| | KCN | 82.15 | 72.69 | 65.38 | 60.17 | 55.68 | 77.38 | 63.37 | 53.05 | 50.09 | 45.47 | 75.32 | 49.85 | 43.49 | 40.61 | 37.98 |
| | KT | 82.15 | 73.02 | 66.36 | 61.56 | 57.23 | 77.38 | 65.89 | 60.06 | 56.68 | 53.40 | 75.32 | 56.95 | 51.68 | 47.59 | 44.08 |
| | EMP | 82.52 | 71.85 | 65.48 | 60.95 | 58.19 | 79.66 | 63.78 | 53.79 | 51.02 | 50.17 | 77.96 | 55.86 | 49.41 | 47.19 | 44.98 |
| | BERT_QA_Arg | 85.00 | 54.65 | 39.26 | 31.73 | 27.95 | 78.63 | 48.14 | 36.72 | 28.14 | 23.81 | 77.36 | 47.17 | 34.79 | 28.16 | 21.98 |
| | ChatGPT | 16.43 | 16.03 | 15.43 | 17.41 | 17.11 | 11.52 | 10.79 | 7.81 | 10.69 | 9.97 | 10.36 | 7.99 | 7.49 | 7.72 | 7.54 |
| | Our model | 84.35 | 80.75 | 76.39 | 76.10 | 73.08 | 80.73 | 68.69 | 63.82 | 59.96 | 58.18 | 78.88 | 68.03 | 61.74 | 57.31 | 55.87 |
| Arg. extract. | Fine-tuning | 48.45 | 35.00 | 23.29 | 18.42 | 17.51 | 50.39 | 30.57 | 22.16 | 17.46 | 14.27 | | | | | |
| | Joint-training | 48.45 | 51.73 | 48.85 | 48.84 | 47.89 | 50.39 | 45.54 | 43.34 | 41.49 | 40.11 | | | | | |
| | BERT_QA_Arg | 50.63 | 35.63 | 25.81 | 20.31 | 19.13 | 48.41 | 29.34 | 22.12 | 17.59 | 13.91 | | | | | |
| | ChatGPT | 4.59 | 2.53 | 5.61 | 5.43 | 5.59 | 3.36 | 4.59 | 3.71 | 3.19 | 3.84 | | | | | |
| | Our model | 48.45 | 47.24 | 42.30 | 40.36 | 37.90 | 50.39 | 39.49 | 35.94 | 34.19 | 31.25 | | | | | |

Table 1: F1 scores of event detection and argument extraction on the ACE05-EN+, ERE-EN, and MAVEN datasets. The best and second-best scores except for joint-training are marked in **bold** and with underline, respectively.

prediction. KCN, KT, and EMP are only feasible for event detection so we cannot apply them to argument extraction. *BERT_QA_Arg* (Du and Cardie, 2020) casts event extraction as a question-answering task. It is capable of transferring specific knowledge based on its question templates. Besides, we consider *ChatGPT*[2] as a zero-shot event extraction model. Following (Li et al., 2023), we use the same prompt in the experiments.

**Evaluation metrics.** We assess the performance of models with F1 scores and report the average F1 scores over all permutations for each model. For event detection, we follow (Cao et al., 2020; Yu et al., 2021; Liu et al., 2022) to calculate F1 scores. For argument extraction, an argument is correctly classified if its event type, offsets, and role label all match the ground truth.

## 5.2 Results and Analyses

### 5.2.1 Main Results

Table 1 shows the main results of event detection and argument extraction. Note that the results on $T_1$ are based on each model's own baseline.

For continual event detection, our model outperforms all competitors on three datasets by a large margin. After training on all tasks, compared with the state-of-the-art models KT and EMP, our model gains 14.89%, 4.78%, and 10.89% improvement of F1 scores on ACE05-EN+, ERE-EN, and MAVEN, respectively. The significant gap demonstrates that semantic confusion on event types is the key to continual event extraction. We can also conclude that our confusion rectification is effectively adaptable to different datasets. Furthermore, the results

[2]https://chat.openai.com/

of our model are even close to joint-training, especially on ACE-05EN+. Differently, our model is cost-effective to handle newly emerging event types without retraining the model on all seen types.

For continual argument extraction, our model also achieves superior performance. This indicates that based on accurately detected event types, our model can extract corresponding arguments well with a few memorized instances. Thus, the performance of continual event extraction highly relies on the effectiveness of continual event detection.

Regarding BERT_QA_Arg, it performs poorly under our setting. This shows that conventional event extraction models may not handle continual event extraction well, although it has transferability on event types. The results of ChatGPT indicate that large language models, which can be viewed as zero-shot event extraction methods, also struggle with continual event extraction.

### 5.2.2 Ablation Study

To validate the effectiveness of each module in our model, we conduct an ablation study on continual event detection, and the results are listed in Table 2. Specifically, for "w/o DA", we use the original dataset rather than the augmented data with pseudo labels; for "w/o AFD", we disable the attention feature distillation module; for "w/o SPD", we disable the selective prediction distillation module; for "w/o PKD", we remove the pivotal knowledge distillation module; for "w/o PKT", we directly train the model without transferring the knowledge of event prototypes to long-tailed event types. We observe that all modules are effective.

| Event detection | | $T_1$ | $T_2$ | $T_3$ | $T_4$ | $T_5$ |
|---|---|---|---|---|---|---|
| ACE205-EN+ | Intact model | 84.35 | 80.75 | 76.39 | 76.10 | 73.08 |
| | w/o DA | 84.35 | 77.58 | 75.18 | 68.98 | 67.19 |
| | w/o AFD | 84.35 | 78.83 | 76.22 | 74.46 | 71.89 |
| | w/o SPD | 84.35 | 78.09 | 73.28 | 68.82 | 65.85 |
| | w/o PKD | 84.35 | 79.16 | 71.01 | 65.70 | 65.06 |
| | w/o PKT | 84.35 | 79.12 | 74.87 | 74.89 | 71.25 |
| ERE-EN | Intact model | 80.73 | 68.69 | 63.82 | 59.96 | 58.18 |
| | w/o DA | 80.73 | 63.28 | 58.10 | 57.54 | 54.12 |
| | w/o AFD | 80.73 | 68.30 | 63.61 | 59.69 | 57.69 |
| | w/o SPD | 80.73 | 67.71 | 61.34 | 56.69 | 55.05 |
| | w/o PKD | 80.73 | 66.83 | 62.04 | 58.50 | 53.83 |
| | w/o PKT | 80.73 | 68.55 | 63.09 | 59.74 | 57.72 |
| MAVEN | Intact model | 78.88 | 68.03 | 61.74 | 57.31 | 55.87 |
| | w/o DA | 78.88 | 64.70 | 53.80 | 52.08 | 48.09 |
| | w/o AFD | 78.88 | 67.29 | 60.78 | 56.77 | 54.96 |
| | w/o SPD | 78.88 | 66.71 | 60.45 | 55.42 | 52.57 |
| | w/o PKD | 78.88 | 65.69 | 59.92 | 54.57 | 51.86 |
| | w/o PKT | 78.88 | 67.76 | 60.68 | 56.74 | 55.20 |

Table 2: F1 scores of ablation study on event detection. All models have the same results on $T_1$ since continual learning has not been executed.

| Event detection | | $T_1$ | $T_2$ | $T_3$ | $T_4$ | $T_5$ |
|---|---|---|---|---|---|---|
| ACE205-EN+ | Fine-tuning | 80.09 | 42.65 | 34.28 | 31.97 | 20.76 |
| | Joint-training | 80.09 | 77.87 | 75.13 | 72.47 | 71.89 |
| | KCN | 73.37 | 70.11 | 63.59 | 55.39 | 52.23 |
| | KT | 73.37 | 71.71 | 64.94 | 58.66 | 54.26 |
| | EMP | 78.35 | 70.57 | 64.58 | 59.50 | 54.72 |
| | BERT_QA_Arg | 81.39 | 54.58 | 41.88 | 32.10 | 24.06 |
| | Our model | 80.09 | 75.36 | 74.20 | 72.24 | 70.45 |
| ERE-EN | Fine-tuning | 76.45 | 47.64 | 36.40 | 24.00 | 22.79 |
| | Joint-training | 76.45 | 70.27 | 65.37 | 64.72 | 61.12 |
| | KCN | 69.34 | 58.27 | 51.87 | 47.02 | 41.73 |
| | KT | 69.34 | 61.57 | 58.88 | 54.16 | 51.54 |
| | EMP | 76.43 | 60.22 | 57.51 | 52.62 | 49.20 |
| | BERT_QA_Arg | 76.21 | 47.68 | 35.87 | 24.98 | 22.76 |
| | Our model | 76.45 | 67.42 | 62.49 | 61.27 | 59.14 |
| MAVEN | Fine-tuning | 75.89 | 47.30 | 33.10 | 24.93 | 22.44 |
| | Joint-training | 75.89 | 67.97 | 64.45 | 63.04 | 62.56 |
| | KCN | 62.61 | 48.09 | 43.46 | 37.41 | 35.06 |
| | KT | 62.61 | 54.71 | 49.55 | 45.63 | 42.21 |
| | EMP | 74.40 | 53.13 | 48.44 | 45.72 | 42,64 |
| | BERT_QA_Arg | 74.23 | 46.21 | 33.72 | 25.58 | 24.27 |
| | Our model | 75.89 | 64.73 | 59.28 | 56.84 | 56.52 |

Table 3: F1 scores of event detection on long-tailed event types. All models are trained with all event types but only evaluated on long-tailed event types.

### 5.2.3 Analysis of Long-Tailed Event Types

We analyze the performance of long-tailed event types in continual event detection and Table 3 presents the results. Note that we do not involve ChatGPT in this setting since we regard it as a zero-shot model that does not suffer from imbalanced data. From the results, our model achieves

| | ACE05-EN+ | ERE-EN | MAVEN |
|---|---|---|---|
| KCN | -28.02 | -36.75 | -33.43 |
| KT | -29.49 | -29.49 | -27.71 |
| EMP | -22.80 | -29.76 | -28.76 |
| Our model | **-18.41** | **-22.64** | **-21.83** |

Table 4: BWT scores of event detection on ACE05-EN+, ERE-EN dataset, and MAVEN.

the best performance on all datasets. Compared with the competing models, our model obtains the lowest performance decline on ACE05-EN+ and even gains improvement on ERE-EN and MAVEN. Thus, our model is effective in saving long-tailed event types from semantic confusion and classifying them correctly.

### 5.2.4 Analysis of Knowledge Transfer Ability

We use a widely-used metric *backward transfer* (BWT) (Lopez-Paz and Ranzato, 2017) to measure the knowledge transfer ability and how well the model alleviates catastrophic forgetting. The BWT score is defined as follows:

$$\text{BWT} = \frac{1}{K-1} \sum_{i=1}^{K-1} \left( \text{F1}_{K,i} - \text{F1}_{i,i} \right), \quad (12)$$

where $K$ is the number of tasks. $\text{F1}_{i,j}$ is the F1 score on the test set of task $T_j$ after training the model on task $T_i$. Note that BWT scores are negative due to catastrophic forgetting. A higher score indicates a better performance.

Table 4 shows the results. Our model performs best, indicating its superiority in alleviating catastrophic forgetting. Benefiting from semantic confusion rectification, our model has better transferability than the competing models.

### 5.2.5 Analysis of Memory Size Influence

The performance of memory-based models is highly related to memory size. We conduct an experiment with different memory sizes. Table 5 lists the results on ACE05-EN+, ERE-EN, and MAVEN. It is observed that our model maintains state-of-the-art performance with different memory sizes. Compared to other models, the performance gap of our model between memory sizes 5 and 20 is the smallest, which demonstrates the robustness of our model to the change of memory size.

## 6 Conclusion

In this paper, we observe the continual learning of event types suffering from semantic confusion

| ACE | Memory size 5 | | | | Memory size 20 | | | |
|---|---|---|---|---|---|---|---|---|
| | $T_2$ | $T_3$ | $T_4$ | $T_5$ | $T_2$ | $T_3$ | $T_4$ | $T_5$ |
| KCN | 71.29 | 63.14 | 58.57 | 52.71 | 72.36 | 65.92 | 63.57 | 59.08 |
| KT | 72.35 | 64.74 | 59.75 | 55.31 | 74.75 | 68.76 | 64.93 | 60.74 |
| EMP | 72.02 | 64.29 | 59.57 | 54.83 | 74.09 | 68.49 | 64.26 | 60.14 |
| Ours | **80.15** | **73.41** | **71.55** | **70.29** | **81.72** | **78.15** | **76.66** | **73.38** |

| ERE | Memory size 5 | | | | Memory size 20 | | | |
|---|---|---|---|---|---|---|---|---|
| | $T_2$ | $T_3$ | $T_4$ | $T_5$ | $T_2$ | $T_3$ | $T_4$ | $T_5$ |
| KCN | 56.60 | 49.70 | 43.11 | 36.64 | 64.65 | 55.10 | 52.32 | 47.14 |
| KT | 63.68 | 58.48 | 53.62 | 47.54 | 66.63 | 60.73 | 58.84 | 55.23 |
| EMP | 62.20 | 54.02 | 50.05 | 46.41 | 66.21 | 56.08 | 55.62 | 53.90 |
| Ours | **69.31** | **61.74** | **58.71** | **54.63** | **68.80** | **64.49** | **62.58** | **59.96** |

| MAVEN | Memory size 5 | | | | Memory size 20 | | | |
|---|---|---|---|---|---|---|---|---|
| | $T_2$ | $T_3$ | $T_4$ | $T_5$ | $T_2$ | $T_3$ | $T_4$ | $T_5$ |
| KCN | 41.51 | 35.97 | 32.01 | 28.36 | 53.18 | 46.10 | 43.84 | 40.93 |
| KT | 50.84 | 45.29 | 40.87 | 37.91 | 59.47 | 53.66 | 49.54 | 47.19 |
| EMP | 49.35 | 43.52 | 40.17 | 39.07 | 59.67 | 52.33 | 50.12 | 48.20 |
| Ours | **68.00** | **60.80** | **56.11** | **54.44** | **68.71** | **62.06** | **59.26** | **57.49** |

Table 5: F1 scores of event detection w.r.t. memory size on ACE05-EN+, ERE-EN, and MAVEN.

and propose a novel continual event extraction model. Specifically, we mark pseudo labels in training data for previously seen types. For newly emerging types, we select accurate knowledge to transfer. For long-tailed types, we enhance their semantic representations by the semantics of associated event types. Experiments on three benchmark datasets show that our model achieves superior performance, especially on long-tailed types. Also, the results verify the effectiveness of our model in alleviating catastrophic forgetting and rectifying semantic confusion. In future work, we plan to study continual few-shot event extraction or other classification-based continual learning tasks.

## Limitations

Our model may have two limitations: (1) It requires an additional memory space to store a few instances, which is sensitive to storage capacity. (2) It relies on the selection of instances in memory space. The prototype knowledge transfer may suffer from the low quality of selected instances in memory space, causing a performance decline.

## Acknowledgments

This work was supported by the National Natural Science Foundation of China (No. 62272219) and the Collaborative Innovation Center of Novel Software Technology & Industrialization.

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

## A Dataset Statistics and Splits

In this section, we introduce how we split the datasets. For ACE05-EN+ (Doddington et al., 2004), we first follow (Lin et al., 2020) to pre-process the dataset. The original development set and test set miss several event types and the number of instances in the test set is much smaller than that in the development set. So we re-split the training set for the development set and the test set following (Yu et al., 2021). We combine the original development set and test set as a new test set. Then, we randomly sample 10% of instances from the training set as a new development set. For the new test set, if the number of instances for a type is less than 10% of the total instances for this type, we randomly sample instances from the training set and remove them from the training set to make up for the difference. We do not include "*Justice:Pardon*" in both the development set and the test set, as it only has 2 instances in the entire dataset. For ERE-EN (Song et al., 2015), we follow the above method to split the dataset as well. For MAVEN (Wang et al., 2020), we take the original development set as the test set and re-split the training set by the above method.

We show the statistics of the new splits in Table 6. Following conventional event extraction (Du and Cardie, 2020; Lin et al., 2020; Wang et al., 2022), we use all unlabeled tokens as negative tokens for the three datasets.

| Datasets | Splits | # Events | # Arguments |
|----------|--------|----------|-------------|
| ACE05-E+ | Training | 3,908 | 5,624 |
|          | Development | 457 | 659 |
|          | Test | 946 | 1,466 |
| ERE-EN   | Training | 5,546 | 7,505 |
|          | Development | 639 | 895 |
|          | Test | 1,095 | 1,501 |
| MAVEN    | Training | 70,112 | - |
|          | Development | 7,881 | - |
|          | Test | 18,904 | - |

Table 6: Statistics of the re-split ACE05-EN+, ERE-EN, and MAVEN datasets.

## B Environment and Hyperparameters

We run all the experiments on an X86 server with two Intel Xeon Gold 6326 CPUs, 512 GB memory, four NVIDIA RTX A6000 GPU cards, and Ubuntu 20.04 LTs. We use a grid search to choose the hyperparameter values. The search space of key hyperparameters is as follows: (1) The search range for the dropout ratio is $[0.1, 0.6]$ with a step size of 0.1. (2) The search range for $\alpha, \beta$ is $[0.1, 2.0]$ with a step size of 0.1. (3) The search range for the $L$ is $[1, 12]$ with a step size of 1. (4) The search range for all learning rates is $[1 \times 10^{-5}, 1 \times 10^{-4}]$ with a step size of $1 \times 10^{-5}$. (5) The search range for the threshold $\tau$ is $[0.65, 0.95]$ with a step size of 0.05. The selected values are listed in Table 7.

| Hyperparameters | Values |
|-----------------|--------|
| Batch size | 8 |
| Dropout ratio | 0.2 |
| $\alpha, \beta, L$ | 1, 1, 3 |
| Gradient accumulation steps | 1 |
| Learning rate for event detection encoder | $5 \times 10^{-5}$ |
| Learning rate for event detection classifier | $5 \times 10^{-5}$ |
| Learning rate for argument extractor | $5 \times 10^{-5}$ |
| Learning rate for entity extractor | $3 \times 10^{-5}$ |
| Dimension of hidden representations | 768 |
| Dimension of feature representations | 512 |
| Threshold $\tau$ for pseudo-labeling | 0.8 |

Table 7: Hyperparameter setting in our model.