# OpenReview forum: "Continual Event Extraction with Semantic Confusion Rectification"
_EMNLP/2023/Conference — EMNLP 2023 Main_

### Official Review · Reviewer_ssjz · 2023-08-01

**Soundness:** 3

**Excitement:**

4: Strong: This paper deepens the understanding of some phenomenon or lowers the barriers to an existing research direction.

**Paper Topic And Main Contributions:**

This paper focuses on continual event extraction, in which event extraction models are consecutively trained on a sequence of different training datasets, each labeled with a different set of event types. When evaluated on each dataset’s corresponding test set, a key concern is that the models not forget the event types they learned earlier, even if those older event types are not labeled in the latest train and test data. The specific tasks the authors focus on are classifying event trigger tokens into event types, and argument tokens into event argument roles.

The authors build on past research in continual learning and specifically continual event extraction. They add several innovations on top of recent state of the art methods. One contribution is to add “pseudo labels” to each new training dataset by applying a model trained on previous datasets to automatically label the previously learned event types in the new data.

Another is to constrain learned features and predictions to not vary far from previous models, but only for tokens assigned to previously learned event types. And another contribution is to identify prototypical feature vectors for long-tail event types, based on the average and standard deviation of feature vectors for observed training examples with that event type, then use those prototypical feature vectors for additional learning.

They authors provide thorough experimental results, comparing their model to relevant baselines and replications of previous state of the art models for this task. They also provide an ablation study to show the added value of each of their specific innovations. Their model performs better than previous approaches overall, on three different event extraction datasets, and each of their main contributions appears to be beneficial to the overall result.

**Reasons To Accept:**

This paper contains multiple innovations in improving continual event extraction. Some of the authors’ contributions are perhaps more incremental tweaks on tops of previous methods, and some seem to be more wholesale additions to their model. Both types of innovations seem to improve their model’s performance, and could be valuable to other researchers working on similar problems.

The authors draw inspiration both from past research on the same task, and on continual learning applied to other tasks. Their contributions might be valuable to others working on continual learning not only in event extraction, but other areas as well. This type of work does seem particularly valuable in event extraction, because EE systems are so reliant on painstakingly labeled data for very specific types of events, and new applications might not share the same event types.

Overall, the methodology appears to be sound and the results are impressive. The ablation study is thorough and helps to address clear questions one might have about what specific revisions or additions to past models contributed most to the improved performance. This break-down should help other researchers considering each of the authors’ contributions as they might be relevant in other settings.

**Reasons To Reject:**

There is a lot going on in this paper, and it is at times hard to follow the exact technical details. I think that arises partly because the authors rely heavily on citations to define many of the methods they’re building on, without always summarizing what exactly they use from other papers. They include some comments about what is different, as in when a key aspect of their approach was not done by past researchers, but I wasn’t always able to follow how much of the authors’ specific method was borrowed from past work.

I think the authors might also overly rely on mathematical formulas (which are welcome) but without always summarizing them in simple, commonly used language. For instance, in more than one place, the authors show a loss function that appears to be cross-entropy loss, but they never use that term. It would be helpful to understand if they are using standard cross-entropy loss, or if there is anything different about their implementation than would be the most common usage, to more easily interpret what they are conveying.

Similarly, the attention feature distillation methodology is hard to follow. Some of the symbols in equation (3) are defined somewhat vaguely. I’m not sure what exactly the output (described as an “attentive feature”) represents. It’s a sum over all tokens of some average of self-attention weights multiplied by the token’s feature representation, but I’m not sure if the attention weights are averaged over the other tokens when the given token is the query or the key. It’s hard to connect the formula to the motivation. Is the goal to place greater constraint on features that came from other tokens that were assigned more weight by a token that was classified as one of the previous event type triggers? If so, why can’t the model simply constrain the learned features of the actual tokens that were predicted to be the event triggers?

At times, I would read through multiple paragraphs unsure what a term meant, until the authors did eventually provide a sufficiently clear definition. For instance, while continual learning might be a growing area of research, the structure of the problem wasn’t even briefly outlined in the Intro or Related Works sections (despite discussion in the latter section about how others have approached it) until the Task Definition in section 3. I felt similarly with some of the terms of for the authors’ main contributions, like pivotal knowledge distillation and prototype knowledge transfer. Simpler, more explicit, high level definitions of concepts earlier on would help to reduce the burden on the reader, rather than relying on niche terms for much of setup, that imply the reader needs to dig through cited works or scour the full paper once just to get the big picture of what’s going on.

Overall, I think the paper’s contributions are quite interesting and intuitive (as far as I understand them), and the results appear to be impressive. But this sometimes difficulty of understanding the exact details makes me uncertain about my interpretation, which makes it harder to enthusiastically recommend the work for publication. I might be missing clear methodological weaknesses, or the paper might confuse other researchers as well, rather than helping to advance their own work.

**Reproducibility:**

3: Could reproduce the results with some difficulty. The settings of parameters are underspecified or subjectively determined; the training/evaluation data are not widely available.

**Reviewer Confidence:**

3: Pretty sure, but there's a chance I missed something. Although I have a good feel for this area in general, I did not carefully check the paper's details, e.g., the math, experimental design, or novelty.

---

> ### Author Rebuttal · Authors · 2023-08-28
>
> Thank you for your time and effort in providing feedback on our paper. We answer your questions below.
>
> **Q1**: They include some comments about what is different, as in when a key aspect of their approach was not done by past researchers, but I wasn’t always able to follow how much of the authors’ specific method was borrowed from past work.
>
> **A1**: Thanks for your comments. In this paper, we focus on continual event extraction. Below we introduce the originality of our model.
>
> To the best of our knowledge, the methods used in our data argumentation module and prototype knowledge transfer module are novel for continual learning. In the data argumentation module, we augment the training data with pseudo labels predicted by the previous model. In the prototype knowledge transfer module, we propose a new method to enrich the semantics of long-tailed types from associated types. We generate the representations of long-tailed types following the Gaussian distribution with the help of the associated event prototypes.
>
> In the pivotal knowledge distillation module, we follow previous works and use the knowledge distillation method as it is a widely used strategy in continual learning. However, we observe that the knowledge distillation method directly used in previous works leads to semantic confusion as we describe in Section 4.2.3. So, we ameliorate it and propose our pivotal knowledge distillation method. Unlike previous works, we expect the model to pay more attention to the tokens associated closely with the event triggers in the attention feature distillation module. In the selective prediction distillation module, we directly leverage the tokens of the previously seen types and "NA" type to transfer knowledge, avoiding confusing the current model with wrong semantics. Furthermore, we do not transfer the probability distribution of "NA" owing to the availability of negative training data on every task.
>
> For the base model and experience reply, we use an event classifier and memory space like previous works for a fair comparison.
>
> **Q2**: I think the authors might also overly rely on mathematical formulas (which are welcome) but without always summarizing them in simple, commonly used language. For instance, in more than one place, the authors show a loss function that appears to be cross-entropy loss, but they never use that term. It would be helpful to understand if they are using standard cross-entropy loss, or if there is anything different about their implementation than would be the most common usage, to more easily interpret what they are conveying.
>
> **A2**: Thank you for pointing this out. We will add easily-understandable summaries of mathematical formulae in the revision. Specifically, Eqs. (2) and (11) in our paper are standard cross-entropy losses.
>
> **Q3**: Similarly, the attention feature distillation methodology is hard to follow. Some of the symbols in equation (3) are defined somewhat vaguely. I’m not sure what exactly the output (described as an “attentive feature”) represents. It’s a sum over all tokens of some average of self-attention weights multiplied by the token’s feature representation, but I’m not sure if the attention weights are averaged over the other tokens when the given token is the query or the key. It’s hard to connect the formula to the motivation. Is the goal to place greater constraint on features that came from other tokens that were assigned more weight by a token that was classified as one of the previous event type triggers? If so, why can’t the model simply constrain the learned features of the actual tokens that were predicted to be the event triggers?
>
> **A3**: Thank you for your question. The attention weights are not averaged over the other tokens. Please allow us to introduce how to generate an attention feature in detail. To obtain an attention feature for a token $x_j$ , we leverage the self-attention score function from BERT to get the attention weight between $x_j$ and each token x in the sentence. We multiply the attention weight and the feature representation for each token x and sum them up. At last, the result divided by the number of tokens in the sentence is the attention feature for the token $x_j$ . Note that the self-attention score function is calculated as the average of the self-attention weights from the last $L$ layers of BERT and we set $L$ as 3 to get the best performance.
>
> We clarify the connection between Eq. (3) and the motivation below. Our motivation is that the tokens associated closely with the event trigger should be assigned higher weights than others. For this purpose, we leverage the self-attention score function to sharpen the feature of associated tokens in this equation. Therefore, in the feature distillation, the current model would pay more attention to the associated tokens and obtain the critical and precise knowledge of these tokens from the previous model to remember the seen event types. Moreover, with the lower attention on the irrelevant tokens, the current model avoids being confused by their irrelevant semantics.
>
> If the model simply constrains the learned features of the actual tokens that are predicted to be the event triggers, the model would be disturbed by the irrelevant tokens and learn inadequate knowledge from the previous model, leading to catastrophic forgetting.
>
> **Q4**: At times, I would read through multiple paragraphs unsure what a term meant, until the authors did eventually provide a sufficiently clear definition. For instance, while continual learning might be a growing area of research, the structure of the problem wasn’t even briefly outlined in the Intro or Related Works sections (despite discussion in the latter section about how others have approached it) until the Task Definition in section 3. I felt similarly with some of the terms of for the authors’ main contributions, like pivotal knowledge distillation and prototype knowledge transfer. Simpler, more explicit, high level definitions of concepts earlier on would help to reduce the burden on the reader, rather than relying on niche terms for much of setup, that imply the reader needs to dig through cited works or scour the full paper once just to get the big picture of what’s going on.
>
> **A4**: We appreciate your suggestions on how to improve our paper. We will add some brief and simple descriptions of continual learning and each module of our methods in the introduction. We will also revise the definitions of concepts in the revision to make them clearer.
>
> Again, we appreciate the reviewers for the positive feedback and constructive suggestions.

---

### Official Review · Reviewer_4NER · 2023-08-04

**Typos Grammar Style And Presentation Improvements:** N/A
**Soundness:** 4

**Excitement:**

4: Strong: This paper deepens the understanding of some phenomenon or lowers the barriers to an existing research direction.

**Missing References:**

N/A

**Paper Topic And Main Contributions:**

In this paper, the authors proposed a method for continual event extraction with semantic confusion rectification. Semantic confusion and imbalance distribution of events in natural language text are the hinderance towards proper and accurate continual event extraction. This paper addresses these issues of event extraction by introducing new method and achieve new SOTA performance on three dataset.

**Questions For The Authors:**

After addressing the previous issues of event extraction, why did the proposed method not able to outperform the joint training method?

**Reasons To Accept:**

This paper presented a new method for event extraction by addressing the previously existing issues.

**Reasons To Reject:**

Though the authors claimed new SOTA performance but still it did not overcome the joint training approach.

**Reproducibility:**

3: Could reproduce the results with some difficulty. The settings of parameters are underspecified or subjectively determined; the training/evaluation data are not widely available.

**Reviewer Confidence:**

4: Quite sure. I tried to check the important points carefully. It's unlikely, though conceivable, that I missed something that should affect my ratings.

---

> ### Author Rebuttal · Authors · 2023-08-28
>
> Thank you for your time and effort in providing feedback on our paper. We answer your question below.
>
> **Q1**: Though the authors claimed new SOTA performance but still it did not overcome the joint training approach. After addressing the previous issues of event extraction, why was the proposed method not able to outperform the joint training method?
>
> **A1**: In this paper, the joint training approach stores all previous training data in the memory and trains the model on all training data for each new task. It simulates the behavior of re-training. In contrast, our model and other memory-based models only store a small number of previous training data in the memory. For example, when the memory size is set to 10, our model only picks and stores 10 training instances for event type "*Attack*" on the ACE05-EN+ dataset. However, the joint training approach stores all 1,144 training instances for event type "*Attack*" and trains the model on them at the next tasks. Given that the joint training approach uses much more training data, it is difficult for our model (and other memory-based models) to outperform it.

---

### Official Review · Reviewer_p7bH · 2023-08-06

**Soundness:** 4

**Excitement:**

4: Strong: This paper deepens the understanding of some phenomenon or lowers the barriers to an existing research direction.

**Paper Topic And Main Contributions:**

This paper is about extending continual learning to event extraction. The authors propose a new continual event extraction model.

**Reasons To Accept:**

The authors explicitly consider semantic confusion on event types. They propose data augmentation with pseudo labels, pivotal knowledge distillation, and prototype knowledge transfer to rectify semantic confusion. Experiments on three benchmark datasets demonstrate that their model establishes a new state-of-the-art baseline with significant improvement and obtains better performance on long-tailed types.

**Reasons To Reject:**

No specific risk to accept.

**Reproducibility:**

3: Could reproduce the results with some difficulty. The settings of parameters are underspecified or subjectively determined; the training/evaluation data are not widely available.

**Reviewer Confidence:**

3: Pretty sure, but there's a chance I missed something. Although I have a good feel for this area in general, I did not carefully check the paper's details, e.g., the math, experimental design, or novelty.

---

> ### Author Rebuttal · Authors · 2023-08-28
>
> Thank you for your time and effort in providing feedback on our paper. We appreciate your acknowledgment of the merits of our work. We are available to reply to any further questions that you may have.

---

### Meta-Review · Area_Chair_CodW · 2023-09-19

**Recommendation:** 4

**Metareview:**

This paper focuses on continual event extraction which sequentially learn different event types to extract. This paper incorporates "pseudo labels" into each fresh training dataset. These labels are generated by employing a model previously trained on earlier datasets to automatically annotate the previously acquired event categories in the new data. This approach helps prevent the model from forgetting previously learned information. Another technique involves constraining the learned features and predictions to remain consistent with past models. Additionally, they make a valuable contribution by identifying characteristic feature vectors for infrequently occurring event types. This is accomplished by computing the average and standard deviation of feature vectors from observed training examples with those specific event types and subsequently employing these prototypical feature vectors for further learning. From my perspective, this is a solid work on continual event extraction. It builds upon some proven techniques from the field of continual learning and validate them within the context of event extraction. As the reviewer mentioned, there is room for improvement in the writing, particularly in the elaboration of technical details.

---

### Decision · Program_Chairs · 2023-10-07

**Decision:**

Accept-Main

**Comment:**

This paper focuses on continual event extraction which sequentially learn different event types to extract. This paper incorporates "pseudo labels" into each fresh training dataset. These labels are generated by employing a model previously trained on earlier datasets to automatically annotate the previously acquired event categories in the new data. This approach helps prevent the model from forgetting previously learned information. Another technique involves constraining the learned features and predictions to remain consistent with past models. Additionally, they make a valuable contribution by identifying characteristic feature vectors for infrequently occurring event types. This is accomplished by computing the average and standard deviation of feature vectors from observed training examples with those specific event types and subsequently employing these prototypical feature vectors for further learning. From my perspective, this is a solid work on continual event extraction. It builds upon some proven techniques from the field of continual learning and validate them within the context of event extraction. As the reviewer mentioned, there is room for improvement in the writing, particularly in the elaboration of technical details.